# Assessing Genetic Variation in Wild and Domesticated Pikeperch Populations: Implications for Conservation and Fish Farming

**DOI:** 10.3390/ani12091178

**Published:** 2022-05-04

**Authors:** Dimitrios Tsaparis, Thomas Lecocq, Dimitrios Kyriakis, Katerina Oikonomaki, Pascal Fontaine, Costas S. Tsigenopoulos

**Affiliations:** 1Institute of Marine Biology, Biotechnology and Aquaculture (IMBBC), Hellenic Centre for Marine Research (HCMR), 71003 Heraklion, Greece; tsaparis@hcmr.gr (D.T.); kyriakds@gmail.com (D.K.); ekonomak@hcmr.gr (K.O.); 2Faculty of Sciences and Technologies, University of Lorraine, INRAE, URAFPA, F-54000 Nancy, France; thomas.lecocq@univ-lorraine.fr (T.L.); p.fontaine@univ-lorraine.fr (P.F.)

**Keywords:** aquaculture, *cyt b*, differentiation, microsatellites, *Sander lucioperca*

## Abstract

**Simple Summary:**

The pikeperch *Sander lucioperca* (Linnaeus, 1758) is an important fish species in the development of European aquaculture. The aquaculture of a fish species can be facilitated by knowing the genetic variability within and among populations. Here, we assessed the genetic background of 8 wild populations along with 13 broodstocks (i.e., from fish farms) of pikeperch through a combination of genetic markers. We underlined that current broodstocks have a genetic diversity similar to wild populations. When focusing on genetic differentiation, we highlight that European pikeperch populations are divided into two groups: one predominantly present in Northern Europe and around the Baltic Sea and another mainly in Central Europe. Broodstocks appear to contain fish of a single origin with only a few exceptions. Ultimately, we have proposed baseline information about genetic diversity of pikeperch along with a genetic tool that can help pikeperch producers manage and improve their fish stock.

**Abstract:**

The pikeperch is a freshwater/brackish water fish species with growing interest for European aquaculture. Wild populations show signs of decline in many areas of the species natural range due to human activities. The comparative evaluation of genetic status in wild and domesticated populations is extremely useful for the future establishment of genetic breeding programs. The main objective of the present study was to assess and compare the genetic variability of 13 domesticated populations from commercial farms and 8 wild populations, developing an efficient microsatellite multiplex tool for genotyping. Partial cytochrome *b* gene sequences were also used to infer phylogeographic relationships. Results show that on average, the domesticated populations do not exhibit significantly lower levels of genetic diversity compared to the wild ones and do not suffer from inbreeding. Nuclear data provide evidence that pikeperch populations in Europe belong to at least two genetically differentiated groups: the first one is predominantly present in Northern Europe and around the Baltic Sea, while the second one comprises populations from Central Europe. In this second group, Hungarian origin populations constitute a differentiated stock that needs special consideration. Aquaculture broodstocks analyzed appear to contain fish of a single origin with only a few exceptions.

## 1. Introduction

The pikeperch (Percidae, *Sander lucioperca*) is a temperate predatory freshwater fish species that tolerates brackish waters [1]. After the end of the last Ice Age, the species spread from the Black–Caspian Sea region to the Aral, Baltic, North, and Aegean Sea basins [2]. From the 19th century, the species was introduced in many surrounding areas such as Anatolia, Central Asia, Siberia, Western Europe, and Western Mediterranean regions [1,3]. This expansion was fostered by both human-made canals connecting ancestral ranges with novel drainages and human-mediated translocations [3]. Currently, several pikeperch populations are declining due to natural habitat destruction and overfishing [4]. Indeed, *S. lucioperca* is particularly prized for its filets by European consumers and anglers [1]. Human demand for pikeperch has risen steadily in recent decades. According to FAO [5], the total pikeperch catches from the wild declined from 50,000 tons in 1950 to only 20,000 tons in 2019 as a result of overfishing [6]. This decline has promoted fish stocking on both its native and non-native ranges [3,7,8] and the development of pikeperch aquaculture [1]. In the past two decades, extensive outdoor farming and intensive indoor production increased with the actual level of RAS (Recirculating Aquaculture System) production estimated between 1000 and 2000 tons in Europe.

Population genetic structure and the phylogeography of *S. lucioperca* have been partially investigated in Europe. In some part of the native range, microsatellite-based analyses highlighted an uneven distribution of genetic diversity as well as genetic differentiation between allopatric populations (i.e., geographic differentiation) [9,10,11,12,13]. In the studied areas where the species has been introduced (i.e., south of France and west of Germany), populations display an unexpected high genetic diversity (i.e., similar to native areas; [3,7]), although it would have been lower due to founder effects and genetic bottleneck often happening during species introduction [14]. This is most likely a consequence of significant gene flow between allopatric populations through individuals’ dispersion or human-made stocking [3,7] as observed in Northern European populations [8,15]. Despite these previous studies, the genetic structure and diversity of pikeperch populations over the species distribution range remain for the most part unknown. This lack of knowledge is even more acute for domesticated stocks (i.e., farmed fish populations) since their genetic specificity and diversity have received far less attention to date [15].

Aquaculture development requires species domestication that shapes genetic specificities in farmed fish stocks [16]. These stocks are often characterized by a lower genetic diversity and a genetic divergence compared to the neighboring wild populations [10,11,17]. The former is due to genetic drift and the founder effect during fish stock establishment and subsequent management. The latter results from (i) the use of non-native allopatric genetically differentiated populations to form fish stocks, (ii) hybridizations in farming systems between allopatric genetically differentiated populations, and/or (iii) genetic alterations in domesticated stocks due to a specific selective regime. Genetic differentiation can also be observed between stocks from different fish farms simply because founding individuals in industrial stocks originate from genetically differentiated wild populations or from stocks experiencing a different selection background.

Assessing the genetic diversity and potential differentiation of wild and farmed *S. lucioperca* is an important prerequisite to further develop pikeperch aquaculture. Knowledge of the genetic variability is key to selective breeding programs to limit deleterious inbreeding depression or highlight genetic pools with remarkable features for farming [18]. Considering a long-term breeding program, it is fundamental to ensure sufficient genetic variation within broodstock populations, as this determines the potential for the selection of desired traits or the adaptation to indoor and inexperienced rearing conditions. Thus, genetic diversity is a valuable piece of information for fish farmers to use to improve and manage their fish stocks [18].

From a conservation point of view, when farmed stocks are genetically distinct from wild neighboring populations, deliberate releases or accidental escapes can lead to issues of outbreeding, genetic homogenization, or competition issues threatening genetic intraspecific variability [19,20,21,22,23]. Therefore, information on the genetic specificity of domesticated stocks can be advantageous for developing management strategies to hamper potential genetic issues in neighboring wild populations (e.g., [24]).

The present study aims at filling the gap in knowledge for pikeperch population genetics by developing a highly informative and efficient microsatellite multiplex tool for the species. We further use this tool with mitochondrial marker information to evaluate and compare the genetic diversity and divergence of domesticated stocks from European commercial farms and wild populations.

## 2. Materials and Methods

### 2.1. Biological Material and DNA Extractions

We sampled 8 wild populations of pikeperch from its native and introduced range and 13 domesticated populations from farms that are using recirculating aquaculture systems (RAS) (Figure 1 and Table 1). Information about the origin of the domesticated fishes was obtained from fish farmers (Table 1). Fin clips were collected without killing the fish and stored in absolute ethanol. Sample providers complied with institutional, national, and international guidelines and regulations as well as the Nagoya protocol to obtain the fish samples. No ethics committee approval was necessary for the collection of fin clips. All fish treatments used for sampling were in accordance with the guidelines of the European Directive (2010/63/EU) on the protection of animals used for scientific purposes. In addition, *S. lucioperca* is neither an endangered species nor a species at risk of extinction according to the IUCN (Red List category: Least Concern). DNA extractions were performed following a standard salt precipitation protocol [25]. In total, DNA was successfully extracted and analyzed from 958 fish samples (Table 1).

### 2.2. Microsatellite Genetic Markers and Population Genetics Analyses

To access the genetic variability of the wild and domesticated populations and perform standard population genetic analyses in pikeperch, we optimized and used two highly informative and efficient multiplex panels (a 4-plex and a 7-plex) of 11 microsatellite loci for genotyping fish samples (Appendix A). Microsatellite loci were selected from a wide range of already developed loci for other percids: PflaL3 and PflaL9 from the yellow perch (*Perca flavescens*, [26], Svi4 and Svi18 from walleye (*Stizostedion vitreum*, [27]), Za024, Za038, Za138, Za144, Za199, Za207, Za237 from the Rhone streber (*Zingel asper*, [28]). All these loci had been previously successfully checked for pikeperch and found highly polymorphic [11,28]. 

Polymerase chain reactions (PCR) were performed in 12.5 μL total volume using a QIAGEN Multiplex PCR Plus Kit with the following cycling conditions: for Multiplex1, an initial denaturation step at 95 °C for 5 min, 35 cycles of 30 s at 95 °C, 90 s at 59 °C, 90 s at 72 °C and a final extension for 30 min at 68 °C; for Multiplex2, an initial denaturation at 95 °C for 3 min, 30 cycles of 15 s at 95 °C, 30 s at 58 °C, 90 s at 72 °C and a final extension for 10 min at 72 °C. For each microsatellite locus, the reverse primer in a PCR mix was fluorescently labelled with dyes (6-FAM, Atto-565, Atto-550 or HEX) that conform to ABI 3730 sequencing technology (Appendix A). PCR products were run on an ABI PRISM 3730 sequencer (Applied Biosystems) along with GeneScan 500 LIZ dye size standard, and raw allele sizes were scored from fluorograms using the STRand software (v.2.4.59 http://www.vgl.ucdavis.edu/STRand, accessed on 15 November 2018). 

MICRO-CHECKER v.2.2.3 [29] was used to detect genotypic errors and test for null alleles. Deviations from Hardy–Weinberg equilibrium (HWE) across all the populations were characterized by estimating inbreeding coefficient F_IS_ values with FSTAT 2.9.4 [30]. In principle, positive F_IS_ values indicate that individuals in a population are more related than expected under a model of random mating, whereas negative F_IS_ values indicate that individuals in a population are less related than expected under a model of random mating. FSTAT 2.9.4 was also used to test for linkage disequilibrium (LD). 

Basic genetic diversity indices such as the mean number of alleles per locus (Na), observed (H_O_) and unbiased expected heterozygosity (uH_E_) were estimated for each population with GENALEX v. 6.5 [31]. To account for variation in sample sizes, we also estimated allelic richness (A_r_) with the program FSTAT 2.9.4 [30]. The differentiation among populations was quantified by pairwise F_ST_ values using the software ARLEQUIN 3.5.1.3 [32]. The test for statistical significance was based on 10,000 permutations.

STRUCTURE 2.3.2 [33] was used to infer the most likely population structure based on microsatellite data of the 21 pikeperch populations. The analysis was performed assuming an admixture model and correlated allele frequencies (default models) without *a priori* population information, using a burn-in period of 250,000 and 1,000,000 subsequent MCMC repeats for each *K* value between 1 and 25. Analysis was replicated 10 times. STRUCTURE runs were implemented on a high-performance computing cluster (IMBBC/HCMR: zorba) using PARASTRUCTURE Perl script [34], and STRUCTURE plots were constructed using DISTRUCT [35]. The most likely number of groups was evaluated using the ΔK criterion [36] as implemented in STRUCTURE HARVESTER 0.9.94 [37] as well as by calculating the posterior probability for each K. 

To visualize broad-scale population genetic structure, we also conducted a discriminant analysis of principal components (DAPC) for all loci and samples using the dapc function in the R package ADEGENET [38]. DAPC is a model-free multivariate statistics-based clustering method that does not rely on a particular population genetics model and is thus free of assumptions about Hardy–Weinberg equilibrium or linkage disequilibrium [38].

### 2.3. Mitochondrial DNA Sequencing, Sequence Diversity and Phylogeographic Analyses

Following a previous pikeperch phylogeographic study by [13], we performed a DNA analysis using cytochrome *b* (*cyt b*) mitochondrial gene as a marker. A fragment of the gene was amplified with PCR using primers L15162 (5′-GCAAGCTTCTACCATGAGGACAAATATC-3′) [39] and H15926 (5′-AAGGGKGGATTTTAACCTCCG-3′) [40]. The 10 μL PCR mix included 20–50 ng of template DNA, 1× Taq buffer, 0.2 μΜ of each primer, 0.2 mM dNTP mix, 1 U of Taq polymerase, 2.5 mM MgCl_2_ and ultrapure water. The PCR cycling protocol consisted of an initial step of 3 min at 95 °C, followed by 30 cycles of 45 s at 94 °C, 45 s at 60 °C, and 1 min at 72 °C and a final extension step at 72 °C for 10 min. Purification of PCR products was performed according to a standard ethanol precipitation protocol, and sequencing reactions were carried out using the BigDye Terminator version 3.1 Cycle Sequencing Kit (Applied Biosystems, Inc.). PCR products were sequenced for both directions (on an ABI PRISM 3730 sequencer (Applied Biosystems)) following the manufacturer’s instructions. 

Derived sequences were edited with MEGA 6.06 [41], re-examined manually by visual inspection of raw fluorogram data and then aligned with ClustalW (as implemented in MEGA 6.06). Finally, all sequences were submitted in GenBank, and unique haplotypes were identified with DAMBE5 [42]. Basic diversity indices such as the number of haplotypes (*nh*), nucleotide diversity (*π*; [43]) and haplotype (gene) diversity (*h*) were estimated for each sampled population using ARLEQUIN 3.5.1.3 [32]. With the same software, we also estimated pairwise Φ_ST_ values among populations and performed AMOVA (analysis of molecular variance) [44] to examine distribution of genetic variation among various hypothetical groups of populations. The test for statistical significance was based on 10,000 permutations. 

To assign our haplotypes to the haplogroups previously identified, we integrated in our data pikeperch *cyt b* sequences already available in GenBank using MEGA 6.06 [41]. All haplotypes resulting from the final multiple alignment (Appendix A) were used to infer geographically meaningful groupings by constructing a haplotype network with the median-joining network method [45] and default settings as implemented in the program PopArt (http://popart.otago.ac.nz).

## 3. Results

### 3.1. Polymorphism of Microsatellites

All microsatellite loci showed relatively high levels of polymorphism exhibiting from 8 (PflaL9) to 23 (Svi4) alleles across the whole data set (Appendix A). The mean number of alleles per locus for the 21 populations ranged from 3.5 to 5.7 (Appendix A). The polymorphism of the microsatellite loci in our study seems to be related to the species these loci first described. The most polymorphic were the loci from *S. vitreum* (Ζvi, 2 loci with average number across the total dataset of 21 alleles and 5.5 alleles per population), followed by those from *Z. asper* (Za, 7 loci with average number of 14.85 alleles and 4.7 alleles per population), and last from *P. flavescens* (Pfla, 2 loci with average number of 8.5 alleles and 3.6 alleles per population). Locus PflaL3 was the only one that showed statistically significant signs of “null alleles” (*p* < 0.05) with high frequencies (>20%) in 5 out of 21 populations and thus excluded from all downstream analyses. No significant linkage disequilibrium between any of the rest of the loci was detected across all populations.

### 3.2. Population Genetics Analyses

Basic population genetics parameters (allelic richness, heterozygosity indices, inbreeding coefficients) were calculated for both wild and domesticated stocks (Table 1, Appendix A). The lowest mean number of alleles (Na) was encountered in domesticated stocks and namely in Denmark-1 (2.6), Denmark-3, Finland-2 (2.8), and Denmark-5 (3.1). The highest number of alleles was also found in domesticated populations: two of them originating from Hungary (Denmark-4: 8.2, Hungary-2: 7.8) and one from Germany (Belgium-1: 7.2). The same pattern (lowest–highest) in domesticated stocks is observed in allelic richness values (Table 1, Appendix A). Concerning the wild stocks, Na ranged between 3.7 in Tunisia and 6.2 in Hungary-1. On average, domesticated stocks exhibited a slightly higher Na (4.75 vs. 4.58) and a slightly lower A_r_ (3.63 vs. 3.78) compared to wild stocks; however, values were not significantly different for both Na and A_r_ after an F-test (Appendix A).

Unbiased expected heterozygosity (uH_E_) ranged from 0.355 in Denmark-3 to 0.726 in Hungary-2 and Belgium-1. Denmark-4 of Hungarian origin showed a comparable value (0.725). The highest value among wild stocks was estimated in Hungary-1, and on average, estimates were slightly higher in wild populations (0.578 vs. 0.557 in domesticated) but again not significantly different after an F-test (Appendix A). Inbreeding coefficient (F_IS_) values varied considerably among populations (from −0.446 to 0.131). They were positive in 48% of the populations (10/21) and negative in the rest (11/21), pointing out both cases of heterozygotes deficit and excess (Table 1). However, negative F_IS_ values were statistically significant in only four populations (Denmark-1, Finland-2, Belgium-1, France-2) indicating deviations from Hardy-Weinberg equilibrium due to heterozygotes excess. 

Differentiation among populations was highlighted by statistically significant pairwise F_ST_ values ranging from 0.01 to 0.484 (Figure 2, Appendix A). Small pairwise F_ST_ values (<10%) were estimated between wild populations of geographic proximity or between wild and domesticated populations from the same origin. A close relationship was observed, for example, between the two domesticated populations of Hungarian origin (Hungary-2 and Denmark-4) and between the wild Hungarian population (Hungary-1) and the domesticated Hungary-2. Low differentiation was also observed in Fennoscandian populations: between the two wild Finnish populations (Finland-3 and Finland-4), between Finland-1 (domesticated) and Finland-3 (wild) and between Finland-4 (wild) and Sweden (domesticated). Lastly, a close relationship was observed between the two populations of Czech origin (wild and domesticated) and between the two French populations (wild and domesticated). In contrast, the highest F_ST_ values (>40%) were mainly observed in pairs of the domesticated population Finland-2 with several wild (Tunisia, Czech) and domesticated populations (Denmark-1,2,3).

Population structure analysis suggested a *K* value of two as the most likely number of existing genetic clusters based on the Δ*Κ* criterion (Appendix A). The first cluster (light blue in Figure 3 for *K* = 2) comprised domesticated populations originating from the Netherlands (Belgium-2), Denmark (Denmark-5), Finland (Finland-1,2) and Sweden as well as wild populations from Finland (Finland-3,4) and will be referred as “Fennoscandian” from now on. The second cluster (orange in Figure 3 for *K* = 2) consisted mainly of populations originating from Hungary (Hungray-1,2, Denmark-4), the Czech Republic (Czech Rep., Denmark-2) and the Netherlands (Denmark-1,3) and will be referred as “Central European”. All the abovementioned populations showed high membership coefficients (Q > 95%) to the corresponding clusters. The rest of the populations showed different levels of admixture (64% < Q < 93%) or even included individuals of both clusters (e.g., France-1).

Two more models of clustering (with *K* = 3 and *K* = 16) exhibited high Δ*Κ* values (Appendix A) and thus could be considered likely to describe the genetic structure of the sampled populations. Moreover, the posterior probability for *K* = 16 was the highest among all *K* (1–25) that were tested with STRUCTURE analysis. In the bar plot for *K* = 3 (Figure 3), the “Fennoscandian” cluster retained its outline (now depicted with yellow), while the “Central European” cluster was further divided into two clusters: the first one (orange) consisted only of the Hungarian origin populations (wild and domesticated), while the second one (light blue) comprised the rest of the populations. If we consider 16 genetic clusters, a higher degree of admixture is generally observed in most populations, but interesting groupings still remained evident. Domesticated populations of Hungarian origin (Hungary-2, Denmark-4) were assigned to the same cluster and showed signs of admixture only with the cluster that wild Hungarian population was assigned to. The same pattern was also observed in Finnish domesticated populations that were both assigned to the same cluster that was also dominant in wild Finnish populations. The Swedish population was assigned to a different cluster but showed signs of admixture only with the Finnish cluster. 

A DAPC scatter plot (Figure 4) indicated the existence of three groups in our dataset. A group consisting of the population of Hungarian origin and the “Fennoscandian” group were clearly segregated along the first axis, while all the remaining populations constituted a third group with an intermediate position.

### 3.3. Cytochrome b Diversity and Phylogeographic Analysis

Mitochondrial DNA polymorphism analysis was based on *cyt b* sequences derived from a subsample of each population due to funding and time shortage. A total of 212 pikeperch individuals were sequenced (Table 1; GenBank accession numbers: ON245529 to ON245740). Sequence length was 571 bp, and after alignment, 5 haplotypes were identified (5 variable sites in total). Both global nucleotide diversity *π* and haplotype diversity *h* were low (*π* = 0.0025, *h* = 0.5444), indicating low levels of genetic diversity (Table 1). Most populations (12) showed a single haplotype for the fish analyzed (*h* = 0), and in only 2 populations (Germany, Poland-1) were more than 2 haplotypes identified. 

Differentiation between sampled populations was in some cases high and statistically significant, as indicated by pairwise Φ_ST_ estimations (Appendix A). Genetic structure at a higher geographical scale was investigated by AMOVA analysis and testing of different alternative hypotheses (Locus by locus AMOVA, Tamura–Nei distance matrix, 10,000 permutations). Four scenarios were checked: three of them were based on groupings of populations following the population clusters identified in microsatellite analysis, and one was based on pairwise Φ_ST_ values (Appendix A). The largest component of variation among groups (FCT) was estimated in the fourth scenario (FCT = 0.8559, *p* < 0.05), which seems to better describe the population division according to polymorphisms in *cyt b*. The first group consisted of Hungarian and German origin populations (wild and domesticated) and the Tunisian wild population, while the second group included all the remaining populations.

For the phylogeographic analysis in *S. lucioperca*, 21 more sequences available in Genbank were added in our alignment (Appendix A, [2,3,13,46,47,48]). Final alignment resulted in eight haplotypes that were used to construct the median-joining haplotype network (Figure 5). Among the haplotypes identified, most were located in more than one site, while only a small part of them were site-specific. The network (Figure 5) supported the existence of two haplogroups (such as AMOVA): one found predominantly in Germany, Hungary, Serbia and Tunisia (haplogroup B) and a second one found mostly in Western and Northern Europe (haplogroup A). The partition of haplogroups in each sampled population of this study is depicted in Figure 3.

## 4. Discussion

### 4.1. Genetic Diversity of Wild and Domesticated Populations

Wild populations analyzed in this study show moderate levels of genetic diversity for microsatellite loci (average Ar:3.8 and uH_E_:0.58) close to values previously referred for other wild populations of pikeperch in Europe. Kánainé Sipos et al. [15], for example, reported values of Ar ranging 3.5–4.0 and H_E_ ranging 0.46–0.59 in three natural water populations from the Danube catchment area. Säisä et al. [11] compared three coastal and five freshwater populations of pikeperch in the northern part of the Baltic Sea drainage basin and reported average values of H_E_ 0.34 (for coastal samples) and 0.42 (for the lake samples) and allelic richness of 3.6 (coastal) and 4.6 (lake samples). Eschbach et al. [3] estimated average values over populations of the native and invaded ranges of the species in Germany and found higher values of genetic diversity in invading populations (Ar:5.0 vs. 4.3; heterozygosity Hs:0.67 vs. 0.60). The aforementioned genetic studies were performed with different sets and numbers of microsatellite loci and in some cases used different estimators to describe genetic diversity levels (e.g., H_E_ instead of uH_E_); thus direct comparison with our results should be made with caution.

Analysis of *cyt b* also revealed low levels of diversity (haplotype and nucleotide) in wild populations (h: 0–0.6 and π: 0–0.0035) with half of them showing no diversity at all. The same pattern of modest mitochondrial genetic diversity of *S. lucioperca* has also been reported in previous studies [2,13] and was attributed to the isolation of the species in a single glacial refugium in the Ponto-Caspian region during the last Ice Age. The experienced genetic bottlenecks and subsequent founder effects lowered species genetic diversity [2].

Among wild populations, those from Tunisia and the Czech Republic exhibited the lowest values for all genetic diversity indices (Na < 3.8, A_r_ < 2.9, uH_E_ < 0.47, *h* = 0, *π* = 0). Both populations also showed signs of heterozygosity deficiency (although not statistically significant). The Tunisian population was introduced from Europe, and the most obvious explanation for low genetic diversity could be the consequences of translocation and subsequent founder effects (see also [49]). The studied Czech population is probably suffering from a recent bottleneck, and this should therefore be taken into account for future conservation and management.

In domesticated populations, allelic richness and heterozygosity varied significantly (A_r_: 2.4–5.5; H_o_: 0.40–0.81; uH_E_: 0.43–0.73), indicating different levels of genetic diversity between farms, probably as a consequence of various stock establishment or subsequent management regimes. Overall, diversity estimates and values of an inbreeding coefficient show that domesticated stocks do not suffer from inbreeding. On the contrary, in three populations, statistically significant deviations from the Hardy–Weinberg equilibrium were observed (Denmark-1, Finland-2, Belgium-1). This was due to heterozygosity and not homozygosity excess. If heterozygosity is higher than expected, an isolate-breaking effect (e.g., the mixing of two previously isolated populations) could be the reason. The Belgium-1 population is probably such a case. It seems that it was created by genetically differentiated stocks as indicated by the admixed origin of most of its individuals in STUCTURE analysis (Figure 3). High number of alleles per locus, high value of allelic richness and the highest value of observed heterozygosity among all populations further support this scenario. In Denmark-1 and Finland-2 in which no sign of admixture is evident (even in *K* = 16), heterozygosity excess could be the result of the small founding population size [50,51]. Low values of allelic richness (2.4–2.6) and number of alleles per locus in both populations are also indicative of this explanation.

### 4.2. Differentiation and Genetic Structure

Our phylogeographic analysis clearly supports the existence of two haplogroups of *cyt b* in pikeperch (Figure 5) in concordance with Kohlman et al. [13]. The first haplogroup, which corresponds to type “A” haplotype of Kohlman et al. [13], is predominantly present in most populations of Europe (Netherlands, Finland, Sweden, France, Czech Rep., Poland) but has been also located [2] in Asian populations (Russia, Azerbaijan, Iran). This distribution provides evidence that the Ponto-Caspian region was the main refugium of pikeperch in Southeastern Europe from where the species recolonized adjacent regions [2]. The second haplogroup, which corresponds to type “B” of Kohlman et al. [13], seems to be geographically more restricted and dominant in the Danube drainage (Germany, Hungary, Serbia). This Danubian lineage of pikeperch has also been demonstrated by Eschbach et al. [3] who hypothesized that the Danube region could have been an additional significant refugium during glacial periods from which recolonization and subsequent development of distinct genetic lineages could have possibly happened. Danubian lineages have been also reported for a range of European freshwater fish ([52] and references therein). Haplogroup B is also present in Poland and Russia, where a natural secondary contact zone of both lineages might exist, and in Tunisia, where pikeperch was introduced from European stocks.

Microsatellite analysis provides a more comprehensive view of the contemporary genetic structure of pikeperch in Europe, which is the outcome not only of ancient vicariant differentiation, but also the result of complex processes that happened during the natural or artificial dispersal of populations and the human manipulation of breeding stocks. Our results provide evidence that European pikeperch populations are divided in at least two genetically differentiated groups: the “Fennoscandian” and the “Central European” (Figure 3 and Figure 4). Wild and farmed Finnish populations and domesticated stocks reported to originate from Denmark and Sweden are clearly assigned to the first genetic group, while Hungarian and Czech origin populations (wild and domesticated) and the French wild population are clearly assigned to the second (Figure 3 and Figure 4).

However, most of the currently analyzed wild populations except for Polish populations that showed significant admixture were assigned using microsatellites to one group. A secondary contact of the two divergent lineages in this region or undocumented human-mediated translocations of allopatric stocks may explain this finding. Some domesticated stocks (e.g., Germany, Belgium-1, Denmark-3) contain fish with admixed genotypes, suggesting that mixing of breeders from different geographic locations has happened in specific fish farms, or the source wild population that was used for breeding programs was already admixed. In the domesticated stock France-1, this practice of fish farmers is even more obvious since individuals that are fully assigned to either genetic cluster (and not admixed) are present at the same time in the population. Three domesticated stocks (Denmark-1, Denmark-3, Belgium 2) originate from the Netherlands. Our results show that individuals or wild populations assigned to different genetic groups should coexist in this region, since Denmark-1 and 3 are clearly assigned to the Central European group, while Belgium-2 is attributed to Fennoscandian one.

The populations of Hungarian origin (both wild and farmed) have a series of characteristics that put them in a key position concerning the conservation and management of European pikeperch. They exhibit high levels of genetic diversity (the highest among populations for some indices). All three populations (Hungary-1,2, Denmark-4) are assigned to the same genetic cluster that seems to be differentiated from the rest of the Central European group in case we accept three clusters as the most likely scenario of genetic clustering (Figure 3). The same pattern of genetic structure is also supported from DAPC analysis that clearly outlines the Hungarian and the Fennoscandian groups from all the rest of the populations (Figure 4). The Hungarian group might be another stock associated with Hungarian lakes as opposed to all other populations (Central European) that probably dispersed through the Danube River west and southward. Wild and domesticated populations of Hungarian origin constitute a significant management unit for the species that should be protected from genetic decay or admixture.

### 4.3. Guideline for Future Pikeperch Aquaculture Development

The multiplex panels of this study provide a valuable tool for the development of pikeperch aquaculture since it can be implemented to optimize the creation of initial fish stock by fish farmers. Indeed, understanding population structure and genetic diversity is a prerequisite for decision making in broodstock management and development of selective breeding programs (e.g., [53,54]). Multiplex panels can serve as a tool to assess (i) potential inbreeding, which can result in inbreeding depression, and (ii) genetic divergence between broodstocks, an important piece of information since crossing and/or comparing differentiation of groups of individuals is useful information in selective breeding programs.

For the studied domesticated pikeperch populations, our results show that current fish stocks do not seem to suffer from inbreeding nor strong genetic diversity loss. This means that (i) current broodstock management methods seem to allow maintaining genetic diversity within stocks, and (ii) current domesticated fish stocks could provide a solid basis for future selective breeding programs.

The present overview of genetic background in wild and domesticated pikeperch populations could be further supported by complementary analyses, including higher sample sizes and stocks originating from a broader distribution area as well as a higher number of genetic markers (such as SNPs) coming from whole genome or reduced representation sequencing methodologies.

Current results already provide valuable insight for fish farmers. First, knowing genetic divergences between domesticated fish stocks paves the way (i) to develop accurate pedigree assignment (e.g., [55]) and (ii) to compare genetic specificities and aquaculture potential (sensu [56]; e.g., growth rate, filet yielding), which can ultimately lead to genome-based selection (e.g., [57]).

Second, fish farmers could consider exchanging breeders between genetically differentiated fish stocks. Indeed, this crossbreeding strategy aims at having progeny with better performances than parents through complementarity of advantages of the two parent biological units and heterosis (i.e., hybrid vigor). Nevertheless, this can result in outbreeding depression (e.g., [58]). Indeed, fish in a given (wild) population/strain possess a particular arrangement of alleles at different loci (coadapted gene complexes). Crossing (hybridization) between the reared strains may potentially lead to a breakdown of these complexes, resulting in reduced fitness.

Third, considering genetic divergences between wild and domesticated populations can potentially improve the prevention of environmental risk near farming locations. Special attention should be paid to the risk of escapees in areas where farmed stocks and wild neighboring populations are genetically distinct to avoid nuisances observed in other fish species (e.g., [59,60]). Within the native distribution range of pikeperch, using local populations or at least populations belonging to the same broader genetic group should thus be favored since, even for indoor aquaculture (e.g., RAS), escapees’ risk cannot be excluded (e.g., [61]).

## Figures and Tables

**Figure 1 animals-12-01178-f001:**
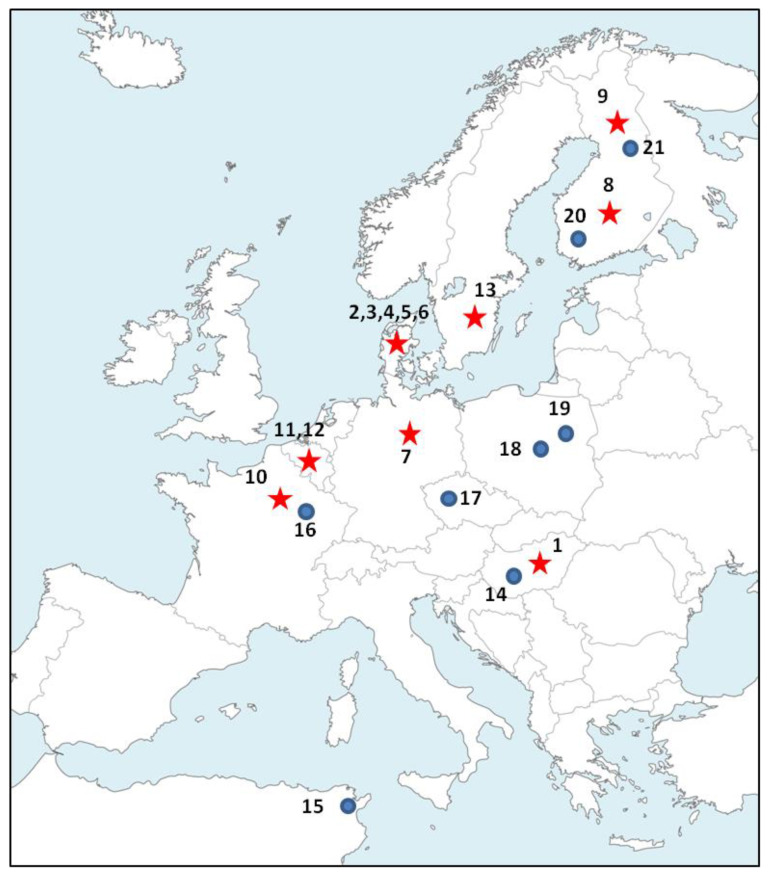
Sampling locations (numbers as in Table 1) of wild (dots) and domesticated populations (stars) of pikeperch (*Sander lucioperca*).

**Figure 2 animals-12-01178-f002:**
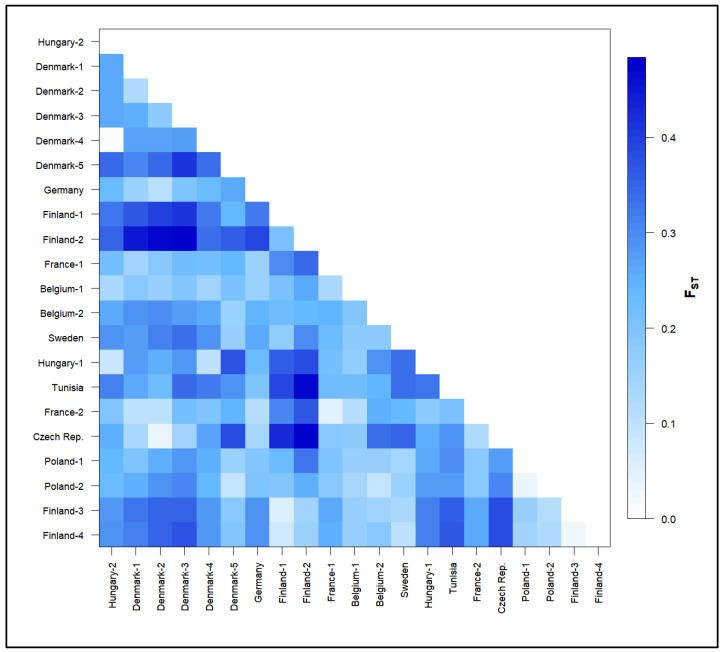
Matrix of pairwise Fst values among pikeperch (*Sander lucioperca*) populations calculated from microsatellite data with ARLEQUIN (distance method: number of different alleles). All values are statistically highly significant (*p* < 0.05).

**Figure 3 animals-12-01178-f003:**
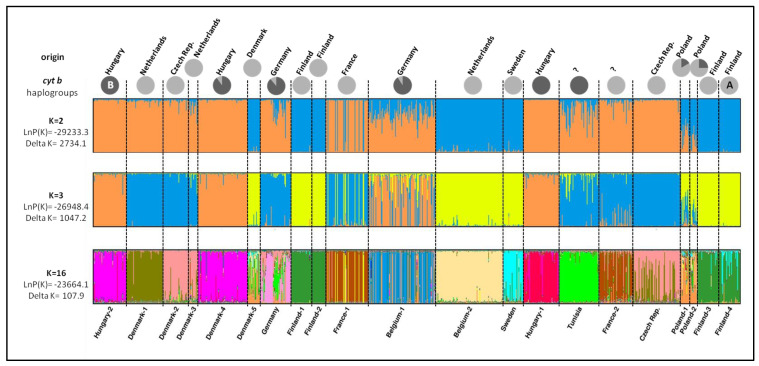
Bayesian individual assignment implemented in STRUCTURE for *K* = 2, 3 and 16 clusters without using geographical area as a priori. The *y*-axis represents the proportions of membership of pikeperch (*Sander lucioperca*) individual genotypes to each *K* inferred cluster. Assignment of each population to *cyt b* haplogroups (A or B) is also depicted with greyscale pie diagrams.

**Figure 4 animals-12-01178-f004:**
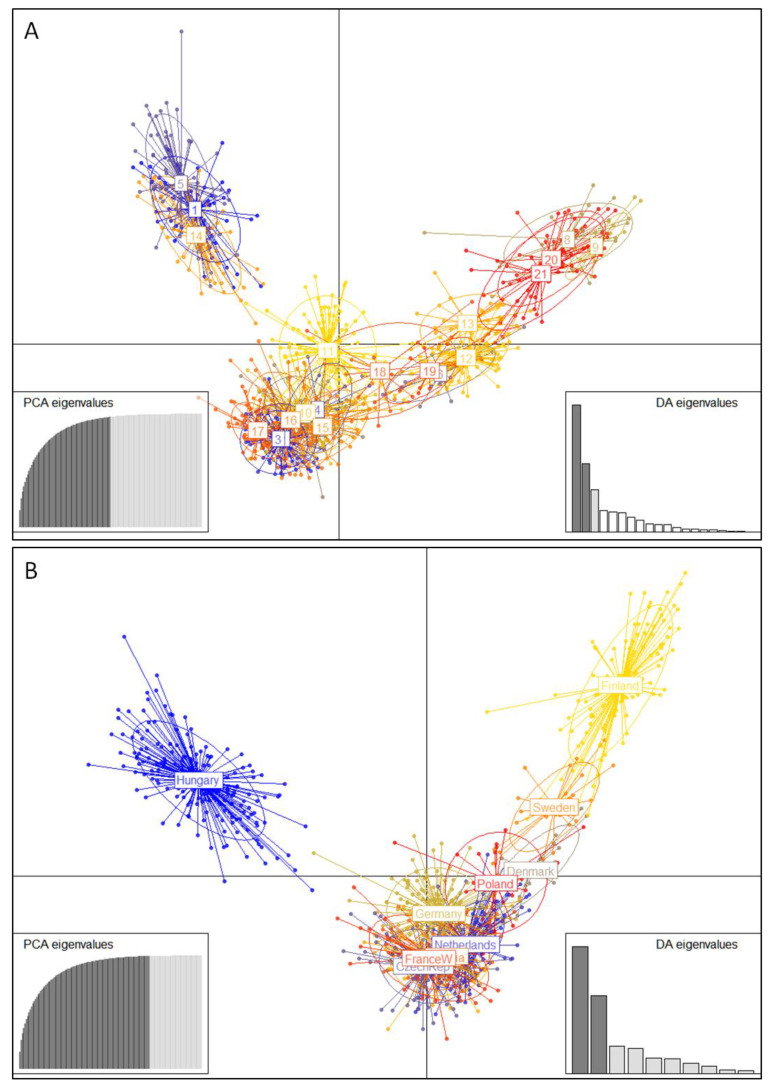
Discriminant analysis of principal components (DAPC) scatter plot. Dots represent individuals of pikeperch (*Sander lucioperca*) with colors denoting sampling population (**A**) or origin (**B**) and inclusion of 95% inertia ellipses. Site numbers in A correspond with Table 1.

**Figure 5 animals-12-01178-f005:**
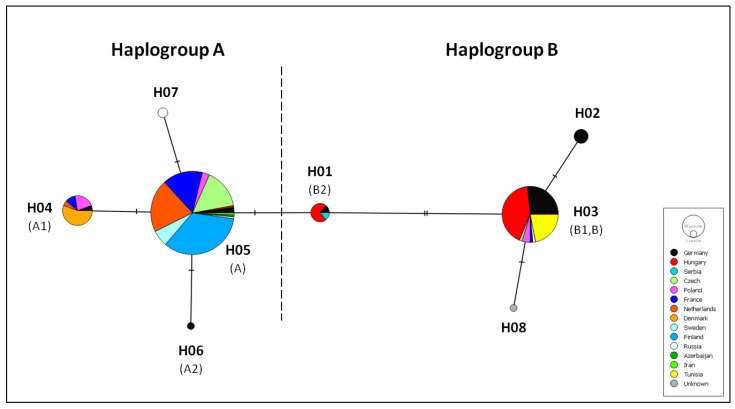
Median-joining haplotype network for *cyt b* (571 bp) of pikeperch (*Sander lucioperca*). Each disc represents a haplotype, and its size is proportional to haplotype frequency. Colors indicate geographic origin. In parentheses are the original names of haplotypes described by Kohlman et al. [13]. The dashed line delimits the two haplogroups based on the analysis of the complete *cyt b* [13].

**Table 1 animals-12-01178-t001:** Basic population genetic parameters of pikeperch (*Sander lucioperca*). For microsatellite loci analysis, the following are indicated: number of samples analyzed (N), mean number of alleles per locus (Na), observed (H_O_) and unbiased expected heterozygosity (uH_E_), allelic richness (A_r_), and inbreeding coefficient (F_IS_). For cytochrome *b* analysis, the following are indicated: number of samples analyzed (*N*), number of haplotypes (*nh*), haplotype (gene) diversity (*h*) and nucleotide diversity (*π*). Asterisks indicate significant deviations from Hardy–Weinberg equilibrium (*p* < 0.05 after Bonferroni correction)**.** Red corresponds to domesticated and blue to wild populations. The symbol int corresponds to populations introduced out of the native species range.

			Microsatellite Analyses							*Cyt b* Analysis			
	Population	Origin	N	Na	H_O_	uH_E_	A_r_	HWE	F_IS_	*N*	*nh*	*h*	*π*%
1	Hungary-2	Hungary	49	7.8	0.676	0.726	5.4	ns	0.070	10	2	0.356	0.125
2	Denmark-1	Netherlands	54	2.6	0.680	0.472	2.4	*	−0.446	10	1	0	0
3	Denmark-2	Czech Rep.	38	3.3	0.488	0.468	2.7	ns	−0.044	11	1	0	0
4	Denmark-3	Netherlands	14	2.8	0.410	0.355	2.6	ns	−0.162	10	1	0	0
5	Denmark-4	Hungary	73	8.2	0.717	0.725	5.5	ns	0.011	10	2	0.200	0.105
6	Denmark-5	Denmark	19	3.1	0.398	0.428	2.7	ns	0.072	10	1	0	0
7	Germany	Germany	46	5.7	0.550	0.563	3.9	ns	0.023	9	3	0.667	0.214
8	Finland-1	Finland	31	3.7	0.582	0.534	3.1	ns	−0.091	11	1	0	0
9	Finland-2	Finland	20	2.8	0.603	0.487	2.6	*	−0.248	10	1	0	0
10	France-1	France	63	5.4	0.591	0.599	3.9	ns	0.013	10	1	0	0
11	Belgium-1	Germany	100	7.2	0.810	0.726	5.1	*	−0.116	11	2	0.182	0.099
12	Belgium-2	Netherlands	100	4.7	0.646	0.619	3.6	ns	−0.045	10	2	0.200	0.035
13	Sweden	Sweden	30	4.4	0.582	0.534	3.6	ns	−0.090	9	1	0	0
14	Hungary-1	Hungary	53	6.2	0.747	0.690	4.9	ns	−0.084	12	2	0.409	0.143
15	Tunisia^int^	Unknown	59	3.7	0.359	0.405	2.7	ns	0.115	14	1	0	0
16	France-2^int^	Unknown	51	4.6	0.671	0.598	4.0	*	−0.122	13	2	0.154	0.027
17	Czech Rep.	Czech Rep.	70	3.8	0.438	0.473	2.9	ns	0.074	10	1	0	0
18	Poland-1	Poland	14	4.6	0.564	0.598	4.2	ns	0.058	6	3	0.600	0.234
19	Poland-2	Poland	11	4.2	0.676	0.644	4.1	ns	−0.052	4	2	0.500	0.350
20	Finland-3	Finland	32	4.8	0.600	0.604	3.6	ns	0.008	11	1	0	0
21	Finland-4	Finland	31	4.7	0.534	0.613	3.9	ns	0.131	11	1	0	0
Total/overall			958	15.4	0.611	0.760		*	0.198	212	5	0.544	0.249

## Data Availability

Not applicable.

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
