# Peer review of "Assessing Genetic Variation in Wild and Domesticated Pikeperch Populations: Implications for Conservation and Fish Farming"

_animals, 2022, doi:10.3390/ani12091178_

Round 1
Reviewer 1 Report
The manuscript by Tsaparis and colleagues entitled “Assessing genetic variation in wild and domesticated pikeperch populations: implications for conservation and fish farming” discusses the genetic diversity of wild and reared populations of Sander lucioperca across Central and North Europe. All manuscript sections are well documented and clearly supported by the scientific literature. The figures and tables are adequate, and the supplementary material is supportive of the results section. The discussion section is robust and provides future perspectives for broodstock management of the species. The manuscript is well written and scientifically sound, therefore it could be published in the journal Animals.
I append only the following two comments:
Line 192: GenBank accession numbers are not referred.
Table 1: the authors should provide an explanation why they didn’t use all the samples used for microsatellite analysis and for cyt b analysis as well. In any case, I don’t believe that the use of cyt b in this kind of analysis is really informative. On the other hand, SSRs are really informative.
Author Response
Reviewer 1
Comments and Suggestions for Authors
The manuscript by Tsaparis and colleagues entitled “Assessing genetic variation in wild and domesticated pikeperch populations: implications for conservation and fish farming” discusses the genetic diversity of wild and reared populations of Sander lucioperca across Central and North Europe. All manuscript sections are well documented and clearly supported by the scientific literature. The figures and tables are adequate, and the supplementary material is supportive of the results section. The discussion section is robust and provides future perspectives for broodstock management of the species. The manuscript is well written and scientifically sound, therefore it could be published in the journal Animals.
I append only the following two comments:
Line 192: GenBank accession numbers are not referred.
Reply: We now provide accession numbers (ON245529-ON245740) in Results (Part 3.3): "A total of 212 pikeperch individuals were sequenced (Table 1; GenBank accession numbers: ON245529 to ON245740)." and more detailed in Table S2
Table 1: the authors should provide an explanation why they didn’t use all the samples used for microsatellite analysis and for cyt b analysis as well. In any case, I don’t believe that the use of cyt b in this kind of analysis is really informative. On the other hand, SSRs are really informative.
Reply: For the sake of reducing costs and due to funding shortage, cyt b analyses were performed just in a random sub-sample (approx. the first 10 fish) of each population genotyped with the microsatellite markers. Therefore, a total of 212 pikeperch individuals were sequenced (Table 1). We added: "Mitochondrial DNA polymorphism analysis was based on cyt b sequences derived from a sub-sample of each population due to funding and time shortage" in Lines 316-317.
Reviewer 2 Report
Dear Authors, I found this manuscript really interesting and of good quality starting from the first reading. Writing is excellent and very clear in all the sections. The experimental study was correctly planned and executed, with proper material and methods and good key results that of course could be very useful for the researchers of this field and not only. Indeed, this manuscript could have a good relapse even from an industrial point of view, in relation to application from aquaculture industry.
I have only two minor suggestion to improve this good research product:
1- please mention Linnaeus 1758 at least the first time you mentioning the specific name Sander lucioperca.
2- double check the bibliography and be sure that all the references were correctly referenced I found a lot of specific names not italicized.
Congrats for this nice work and good luck for the future.
Best regards
The Reviewer
Author Response
Reviewer 2
Comments and Suggestions for Authors
Dear Authors, I found this manuscript really interesting and of good quality starting from the first reading. Writing is excellent and very clear in all the sections. The experimental study was correctly planned and executed, with proper material and methods and good key results that of course could be very useful for the researchers of this field and not only. Indeed, this manuscript could have a good relapse even from an industrial point of view, in relation to application from aquaculture industry.
I have only two minor suggestion to improve this good research product:
1- please mention Linnaeus 1758 at least the first time you mentioning the specific name Sander lucioperca.
Reply: We have now added the reference in the Simple Summary (Line 13)
2- double check the bibliography and be sure that all the references were correctly referenced I found a lot of specific names not italicized.
Reply: We apologize for this; we have now corrected all species names in italics.
Congrats for this nice work and good luck for the future.
Reply: We sincerely thank you for your nice words!
Reviewer 3 Report
There is useful and valuable information here, but this study is not particularly novel.
One cannot infer inbreeding from microsatellite data as you do not know the starting genotype frequencies. The F values generated are no true inbreeding values but demonstrate the overall relatedness and isolation of the populations.
Alternative explanation of the high genetic diversity of the transplanted population………not only number of founders is important but also their initial variability. Theoretically, if they are highly diverse, 10 founders could generate more genetic diversity than 10,000 founders of low diversity. Of course, the explanation offered by the authors is the most likely one, but one must offer all alternatives if there is not 100% conclusive evidence for a single explanation.
The authors have tunnel vision and pick choose hypotheses--- for eample" The former is due to genetic drift and founder effect during fish stock establishment and subsequent management”. This is not the most likely explanation, a more likely explanation is that --- Selective forces are often one of the main if not main factor in causing the different gene frequencies between domestic and wild fish.
The authors overestimate the value of their data. Microsatellites are not necessary for the farmers to have a selective breeding program. In fact, usually most of them are neutral for performance traits. What is critical is the measurement of quantitative genetic parameters. Additionally, microsatellites can sometimes but not always be useful for marker assisted selection. If you want to apply DNA markers, SNPs are much more powerful for genomic selection.
All table and figures should stand alone. If it were separate from the manuscript, it should be understandable. Thus, each legend should minimally mention the common and scientific name of the study organism.
Again, microsatellites are not a prerequisite for selective breeding.
Several false statements are made. Microsatellite variation is not a cornerstone for selective breeding. Microsatellites are an indicator of homozygosity but not inbreeding-------I do not think the authors understand inbreeding----if there was an increase and excess of homozygotes that is a potential indicator of inbreeding but not overall variability.
Statements on lines 470-473 are far overreaching.
In these concluding paragraphs , the authors try to synthesize selective breeding and conservation together and it comes out jumbled. One cannot predict heterosis with microsatellites at all. Quantitative genetics and combining abilities, dominance variation are key.
The final recommendations certainly restrict what genetic advantages could be gained if these guidelines are followed. Prevention of escapement seems like the best option unless you want to potentially prevent the use of the best domestic or selected genotypes.
Many beautiful figures and tables were constructed, and I compliment the authors, but the complexity makes interpretation and understanding slow.
Some old analyses are still quite valuable. Much could be learned , quickly visualized and understood from an old fashioned dendrogram.bb
Author Response
Reviewer 3
Comments and Suggestions for Authors
There is useful and valuable information here, but this study is not particularly novel.
One cannot infer inbreeding from microsatellite data as you do not know the starting genotype frequencies. The F values generated are no true inbreeding values but demonstrate the overall relatedness and isolation of the populations.
Alternative explanation of the high genetic diversity of the transplanted population………not only number of founders is important but also their initial variability. Theoretically, if they are highly diverse, 10 founders could generate more genetic diversity than 10,000 founders of low diversity. Of course, the explanation offered by the authors is the most likely one, but one must offer all alternatives if there is not 100% conclusive evidence for a single explanation.
The authors have tunnel vision and pick choose hypotheses--- for eample" The former is due to genetic drift and founder effect during fish stock establishment and subsequent management”. This is not the most likely explanation, a more likely explanation is that --- Selective forces are often one of the main if not main factor in causing the different gene frequencies between domestic and wild fish.
Reply: We agree with the reviewer’s first statement that the initial variability is also dependent on the number of wild fish included into the breeding nucleus. The closest estimate to variability found in the wild is to analyze “true” wild populations, specifically those sampled in the species natural distribution like in Hungary, Czech Rep., Poland, and Finland. The use of the Fis as an index to estimate inbreeding is generally used in population genetics studies and this is not a novel approach in this study.
For the second one, we regret to say that the wording used “Tunnel vision and pick/choose hypotheses (that obviously are more favorable to us)” is at least not acceptable from a fellow colleague. Unless, he has specific suggestions to propose, we do not accept these accusations.
Based on populations genetics studies and theories as well as the common breeding practices in pikeperch, we insist on hierarchizing first genetic drift and founder effect as the principal causes for the loss of genetic variability. Selective forces are indeed a factor influencing the allele frequencies but at a subsequent level and after few generations of selection and domestication. In most cases, pikeperch broodstocks are collected directly from the wild as eggs or small fish (mainly in Scandinavia) and in fact very recently the purchase of fingerlings from other companies came in practice. Pikeperch aquaculture readiness level cannot be compared to salmon or even the gilthead sea bream, European sea bass and turbot species in which we encounter at least 6-8 generations or 50 years of selection in salmon!
The authors overestimate the value of their data. Microsatellites are not necessary for the farmers to have a selective breeding program. In fact, usually most of them are neutral for performance traits. What is critical is the measurement of quantitative genetic parameters. Additionally, microsatellites can sometimes but not always be useful for marker assisted selection. If you want to apply DNA markers, SNPs are much more powerful for genomic selection.
Reply: We regret to say that we disagree with the reviewer. Genetic markers, microsatellites in our case, are of great help to estimate a series of indexes and phenomena as the heterozygosity and allelic richness of a broodstock, inbreeding (measured as Fis index), dominance in spawning (if some fish contribute more or less in the offspring), and correct paternities are a cornerstone for a successful breeding programme. If someone chooses only the fish with the best attributes (high weigh, FCR, disease resistance, body shape etc), he/she is seriously risking to choose sibs for the next generations. Indeed, microsatellites are (most of them) neutral but SNPs can also be unless they are chosen based on the genic position. We agree that SNPs are more powerful for MAS or genomic selection but obviously this implies that a company wishes to perform this kind of approach through GWAS etc and invest seriously into this direction. Otherwise, a powerful set of microsatellite markers can be used to assess the genetic variability status of a broodstock and this is what we want to emphasize when stating “understanding population structure and genetic diversity is a prerequisite for decision-making in broodstock management and development of selective breeding programs” in lines 462-466. We do not say that microsatellites are needed to establish a selective breeding programme but their use to estimate basic population indices is still a powerful tool.
All table and figures should stand alone. If it were separate from the manuscript, it should be understandable. Thus, each legend should minimally mention the common and scientific name of the study organism.
Reply: We added this information in all legends.
Again, microsatellites are not a prerequisite for selective breeding.
Reply: We have referred to this above. Microsatellites can be used successfully at the very beginning to genetically characterize a broodstock. If this is found highly inbred with a powerful genetic tool (microsatellites or SNPs), we consider that it may be pointless to spend thousands or millions of euros/dollars for selective breeding with any marker.
Several false statements are made. Microsatellite variation is not a cornerstone for selective breeding. Microsatellites are an indicator of homozygosity but not inbreeding-------I do not think the authors understand inbreeding----if there was an increase and excess of homozygotes that is a potential indicator of inbreeding but not overall variability.
Reply: We agree that microsatellite variation might not be a cornerstone for selective breeding and we are reducing its importance in the text (line 467). Regardless of his/her reviewing judgement, the use of microsatellites to estimate inbreeding is still widely used and we may provide a series of recent papers to show to him/her how this can be performed.
Statements on lines 470-473 are far overreaching.
Reply: Thank you, we may agree with that but we prefer to keep them in the final part since we give a conclusion kind summary with associated references.
In these concluding paragraphs , the authors try to synthesize selective breeding and conservation together and it comes out jumbled. One cannot predict heterosis with microsatellites at all. Quantitative genetics and combining abilities, dominance variation are key.
The final recommendations certainly restrict what genetic advantages could be gained if these guidelines are followed. Prevention of escapement seems like the best option unless you want to potentially prevent the use of the best domestic or selected genotypes.
Reply: We are not basing our recommendations based on the microsatellite analyses, and the general guidelines are based on recent publications and research efforts of the partners associated in the current paper.
Many beautiful figures and tables were constructed, and I compliment the authors, but the complexity makes interpretation and understanding slow.
Some old analyses are still quite valuable. Much could be learned , quickly visualized and understood from an old fashioned dendrogram.bb
Reply: Thank you for the comment but we sincerely think that the inclusion of a dendrogram will not facilitate future readers.
Reviewer 4 Report
This manuscript presents a study of wild and farmed populations of Sander lucioperca, a species in expansion in European aquaculture. Microsatellite loci and a fragment of Cyt b sequence were used as genetic markers to build on population genetics of this species at European level and to evaluate possible effects of domestication at the level of population variation. Sample sizes are a little bit small in some cases (Denmark-3, Poland-1 and Poland-2), but are generally OK. The genetic analysis and statistics are sound and the results are well interpreted. This is a correct study that could be improved to gain relevance for Animals journal. A few ideas follow:
- The importance of the species for European aquaculture could be emphasized to attract the attention of generalist readers. I encourage the authors to include more details (briefly in the Introduction, with more details as a first section in Material & Methods) about the production of this species, from extractive fisheries and from aquaculture. FAO reports or any other information source could be used for this.
- A few words about the limitations of this study could be added in the Discussion. The authors could acknowledge limited sample size of several samples, and suggest the use of alternative markers, perhaps whole-genome analysis.
- Sander lucioperca is considered an invasive species in some regions where it was introduced. See for example CABI website https://www.cabi.org/isc/datasheet/65338 The conditions required to farm a non-native species in European settings should be considered in the Discussion.
Author Response
Reviewer 4
Comments and Suggestions for Authors
This manuscript presents a study of wild and farmed populations of Sander lucioperca, a species in expansion in European aquaculture. Microsatellite loci and a fragment of Cyt b sequence were used as genetic markers to build on population genetics of this species at European level and to evaluate possible effects of domestication at the level of population variation. Sample sizes are a little bit small in some cases (Denmark-3, Poland-1 and Poland-2), but are generally OK. The genetic analysis and statistics are sound and the results are well interpreted. This is a correct study that could be improved to gain relevance for Animals journal. A few ideas follow:
- The importance of the species for European aquaculture could be emphasized to attract the attention of generalist readers. I encourage the authors to include more details (briefly in the Introduction, with more details as a first section in Material & Methods) about the production of this species, from extractive fisheries and from aquaculture. FAO reports or any other information source could be used for this.
Reply: We agree with this suggestion and we have added few lines describing the wild capture and aquaculture production in the first paragraph of the Introduction section.
- A few words about the limitations of this study could be added in the Discussion. The authors could acknowledge limited sample size of several samples, and suggest the use of alternative markers, perhaps whole-genome analysis.
Reply: We have now included this in Lines 471-476
- Sander lucioperca is considered an invasive species in some regions where it was introduced. See for example CABI website https://www.cabi.org/isc/datasheet/65338 The conditions required to farm a non-native species in European settings should be considered in the Discussion.
Reply: The RAS production is mainly promoted in areas where pikeperch is present in the wild ecosystems. Moreover, its production is only targeted to human consumption with all fish killed before their sale (as whole fish or filet). This means that its potential impact on native populations of local ecosystems is very low. The problem related to potential negative effect on wild population is linked to fish restocking practices and pond culture (live fish) which is not our subject here. RAS and pond culture are very different for fish escapees.